# Quantitative Sensory Testing in Fibromyalgia Syndrome: A Scoping Review

**DOI:** 10.3390/biomedicines13040988

**Published:** 2025-04-17

**Authors:** Adriana Munhoz Carneiro, Marina de Góes Salvetti, Camila Squarzoni Dale, Valquíria Aparecida da Silva

**Affiliations:** 1Mood Disorders Department-Pro Gruda, Institute of Psychiatry, Faculty of Medicine, University of São Paulo, São Paulo 05403-010, Brazil; adrianacarneiro01@gmail.com; 2Interdisciplinary Neuromodulation Service, Psychiatry Department, School of Medicine, University of São Paulo, São Paulo 05403-010, Brazil; 3Medical-Surgical Nursing Department, School of Nursing, University of São Paulo, São Paulo 05403-000, Brazil; mgsalvetti@usp.br; 4Department of Anatomy, Institute of Biomedical Sciences, University of São Paulo, São Paulo 05508-000, Brazil; camila.dale@usp.br

**Keywords:** quantitative sensory testing, fibromyalgia, pain measurement, CPM, temporal summation, sensory function

## Abstract

**Background/Objectives**: Quantitative sensory testing (QST) is one of the most reliable methods for assessing Fibromyalgia Syndrome (FMS). Despite its importance, there are still controversies regarding the correct interpretation of evoked responses, as they may vary depending on the protocol, individual characteristics, disease severity, and other factors. This study aims to examine how QST has been applied as an outcome measure in FMS. **Methods**: We considered three databases (Medline, Embase, and Web of Science) until June 2024. From a total of 2512 studies, 126 (39 RCTs and 87 non-RCTs) were selected for full reading after assessment for risk of bias and eligibility criteria. These criteria included at least one type of QST and a clear diagnosis of fibromyalgia (FMS). **Results**: The results highlighted a lack of standardization in QST, as no reported protocols were followed and there was no specific number of tender points tested for FMS. Additionally, there was inconsistency in the selection of sites and types of tests conducted. **Conclusions**: This heterogeneity in methodology may affect the comparability and interpretation of results, underscoring the urgent need for standardized guidelines for conducting QST in fibromyalgia studies. A clear understanding of how QST has been measured could prompt a reevaluation of current approaches to FMS assessment, leading to more accurate interpretations and, ultimately, improved management of this complex condition.

## 1. Introduction

Characterized as a widespread chronic pain syndrome, Fibromyalgia Syndrome (FMS) has a complex multifactorial etiopathogenesis that remains not fully understood [1,2] and affects 3 to 6% of global population [3]. FMS is often associated with impairments in mental health and quality of life [4,5,6,7,8,9].

Since 1980, various FMS diagnostic criteria have been developed to reduce subjective clinical judgment, most notably the American College of Rheumatology (ACR), which consider FMS diagnosis as a combined score of the Widespread Pain Index (WPI) and Symptom Severity Scale (SSS) [5]. In this context, Quantitative Sensory Testing (QST) has emerged to improve the precision of sensory deficit detection in FMS by assessing pain thresholds through a combination of static and dynamic protocols that allows the assessment of pain thresholds through isolated stimuli, measuring hyperalgesia or hypoalgesia in specific areas and the perception of pain [10,11].

QST is based on measurements of responses to calibrated, graded innocuous, or noxious stimuli (generally mechanical or thermal) [12,13,14]. Despite its potential, its implementation can be complex due to cost and protocol selection [15,16]. In FMS, QST protocol variability, combined with individual differences and comorbidities, can hinder the interpretation of evoked responses [17,18]. This lack of standardization impedes understanding, comparison of studies, and development of effective diagnostic and therapeutic strategies [14].

Equivalent difficulties have been observed in other chronic pain conditions, already postulated for previous reviews [18,19]. In brief, the use of QST for painful experiences demonstrated the need for a more standardized approach [18,19]. QST standardization issues, including test site variability and inconsistent definitions, have been reported in other chronic pain conditions like knee osteoarthritis [20] and pediatric populations [21], highlighting the need for consistent protocols to improve reproducibility and clinical applicability [21]. This heterogeneity compromises QST’s potential in chronic pain research, including fibromyalgia [22].

Therefore, despite favorable evidence for QST application, there are no previous studies debating the implications of QST protocols in FMS. In this sense, the current scoping review aims to clarify the complexities and variations inherent in QST methodologies in FMS. By examining QST protocols and identifying factors that influence their reliability as outcome measures, we believe it will be possible to develop more effective approaches for fibromyalgia syndrome (FMS).

## 2. Materials and Methods

This scoping review protocol was developed based on the Preferred Reporting Items for Systematic Reviews and Meta-Analyses Extension for Scoping Reviews, PRISMA-ScR [23] (Appendix A). The protocol was previously registered [24].

### 2.1. Eligibility Criteria

This study considered studies where participants were adults, both sexes, aged 18+ with a clear diagnosis of Fibromyalgia Syndrome (FMS) considering any of the ACR criteria. As we are looking for Quantitative Sensory Testing-QST measurement, studies with at least one measure of pain threshold or sensitivity (any study design), were selected. We excluded studies if they had less than 50% of the participants with FMS. Duplicates, reviews, and commentaries on findings from other studies or documents that were not the primary research (for example, conference abstracts) were also excluded.

### 2.2. Search

This review extracted studies from the following databases: Pubmed (*n* = 508), Embase (*n* = 817), and Web of Science (*n* = 1187). We replicated the primary database search terms (Pubmed) for the others (see Appendix A). The search was not limited by language or year. The search was conducted up to 3 June 2024.

### 2.3. Selection of Sources of Evidence and Critical Appraisal

The quality of the included studies was assessed by two independent reviewers (AMC and VAS), and disagreements will be solved by a third reviewer (MGS). For the RCTs, the ROB 2 tool was used, and for the non-RCTs, the STROBE (see Appendix A).

### 2.4. Data Charting Process

The Rayyan software (https://www.rayyan.ai/, accessed on 4 April 2025) [25] was used to select studies by title and abstract. It was made by two independent reviewers (AMC and VAS) based on our previously established inclusion/exclusion criteria. The third reviewer (MGS) remained on standby if needed. After a full reading, data were extracted from papers by AMC and VAS using an extraction table developed by the reviewers independently. In cases where it is not possible, we search for the data protocol or other similar studies made by the author/group in order to clarify the information. In our study, it was not necessary to consult the authors.

### 2.5. Data Items and Synthesis of Results

For extraction, we considered the study design, quality of the study, sample characteristics, age, sex, distribution, inclusion criteria, diagnosis, type of QST (static or dynamic), and methods applied by the studies. We also obtained information from other measurements and main conclusions. The data were tabulated and presented in a narrative way, answering the research objectives.

## 3. Results

The search (up to June 2024) yielded 2512 records: 1187 from Web of Science, 817 from Embase, and 508 from PubMed. No filters (e.g., article type, species, language, or age) were applied during the search process to avoid unintentionally excluding relevant records. The search query used for each database is detailed in Appendix A. The study selection process is summarized in Figure 1. The characteristics of the included studies are summarized in Table 1.

### 3.1. Quality Assessment

Only studies classified as having a low risk of bias or some concerns were considered, and 30 (76.9%) RCTs and 59 (67.81%) non RCT’s studies met this criterion. RCT limitations involved lack of information regarding the original protocol and data analysis plan (Appendix A). In terms of nonRCTs, limitations were related to sample (i.e., recruitment, inclusion, sample size calculation), generalizability and insufficient information regarding sample size and bias (Appendix A).

### 3.2. Narrative Synthesis of Quantitative Sensory Testing Methods

A total of 42.9% of studies included both static and dynamic QST assessments, offering a comprehensive approach to sensory evaluation. (Table 2). For these studies, we divided our results considering both assessments.

### 3.3. Static QST

Static QST methods comprised 76% of all assessments. Pressure pain thresholds/tolerances (PPTh/PPT) were the most frequently measured (*n* = 84), predominantly at 18 tender points (*n* = 24), hands (*n* = 17), trapezius (*n* = 8), and forearm (*n* = 5), using primarily the Somac algometer (*n* = 54). Mechanical detection/pain thresholds/sensitivity (MDTh/MPTh/MPS, *n* = 10) were assessed mainly at the forearm (*n* = 3) and hands (*n* = 4), often with Von Frey monofilaments (*n* = 5). Thermal pain thresholds/tolerances (TPTh/TPT, *n* = 36) were typically measured at hands (*n* = 9) and forearm (*n* = 7), often with the TSA II Medoc.

### 3.4. Dynamic QST

Dynamic QST methods constituted 24% of assessments. Temporal Summation (TS, *n* = 15) was primarily assessed at hands (*n* = 7) and forearm (*n* = 4). Conditioned pain modulation (CPM, *n* = 26) was frequently tested at the forearm (*n* = 12) and hands (*n* = 4), using PPT as the test stimulus and cold water immersion as the conditioned stimulus.

## 4. Discussion

This scoping review examined how QST is used in FMS, a complex condition with widespread pain and variable symptom presentation, that per se makes diagnosis and measurement challenging [15,16]. While QST offers a potential surrogate measure to improve pain assessment reliability and validity, and understand neuropathic pain [4,149], this review revealed important methodological issues.

Although static QST is prevalent, variations in body location, stimulus duration, and intensity may affect results. Researchers demonstrated PPT and CPM variability across test points, reflecting altered pain modulation in FMS. Given the diffuse pain and altered sensation characteristic of FMS, QST at remote sites may reinforce information regarding the central nervous system [18,45,104]. The NeuPSIG consensus [14] reinforces the use of multiple test sites, or preferably standardizing test locations in order to improve the accuracy and interpretation by reducing variability and potentially revealing more consistent patterns of somatosensory dysfunction in FMS.

Given the scarcity of studies measuring QST both before and after interventions, QST stability in FMS remains poorly understood. No consistent information regarding the presence of other symptoms was controlled (e.g., sleep disturbances and other non-physical symptoms), neither the impact of psychological factors, the presence of psychopathology or neuropathic pain conditions [14,18,20]. There is an amount of literature available claiming for a more controlled information of those variables, once they are related to the severity of this disease [5,7]. Further research should explore how internal and external factors contribute to FMS progression [110]. Consequently, despite efforts to minimize bias, the generalizability of findings remains limited due to substantial methodological variation.

The diversity in test locations—from high muscle areas (e.g., trapezius, deltoid) to minimal muscle sites (e.g., wrist, thumbnail)—further adds to this heterogeneity [150]. This inconsistency could limit the synthesis of findings across studies and impact the reliability of QST as a biomarker in FMS. Standardization in test locations and stimuli parameters could facilitate future meta-analyses and enhance the clinical applicability of QST.

Furthermore, QST modality definitions might be implicit. For example, while TS and CPM are often used as measures of central sensitization, they also involve the peripheral nervous system–the parameters of the sensitization analysis must be defined to each study. Additional limitations include variability in test parameters (number of tests, duration, rest intervals, stimulus intensity/increment) and equipment. Despite recommended protocols, application remains uncommon, hindering full standardization in this review.

These factors include variations in body location of QST application, differences in stimulus parameters (e.g., duration, intensity, rest intervals), test modality definitions (e.g., TS and CPM interpretation), and equipment used. Each of these methodological differences can introduce heterogeneity between studies. Although standardized protocols have been recommended, our findings show that their consistent application remains uncommon. In our review, even when controlling this information, it was not possible to standardize fully across studies.

Finally, as a strength, this is one of the first studies to recruit the state of the art by considering QST measures in FMS. We hope that this scoping review might be able to summarize the need for a more standardized approach to measuring FMS, particularly considering the complex and unpredictable nature of endogenous pain inhibition mechanisms.

## 5. Conclusions

While promising for FMS assessment, QST’s potential is hampered by significant lack of information regarding its validity and reliability. Future research should stratify studies by treatment modality (e.g., pharmacological vs. neuromodulatory) to elucidate treatment-specific effects and optimize patient care. Addressing these gaps promises to significantly advance FMS understanding and improve patient outcomes.

## Figures and Tables

**Figure 1 biomedicines-13-00988-f001:**
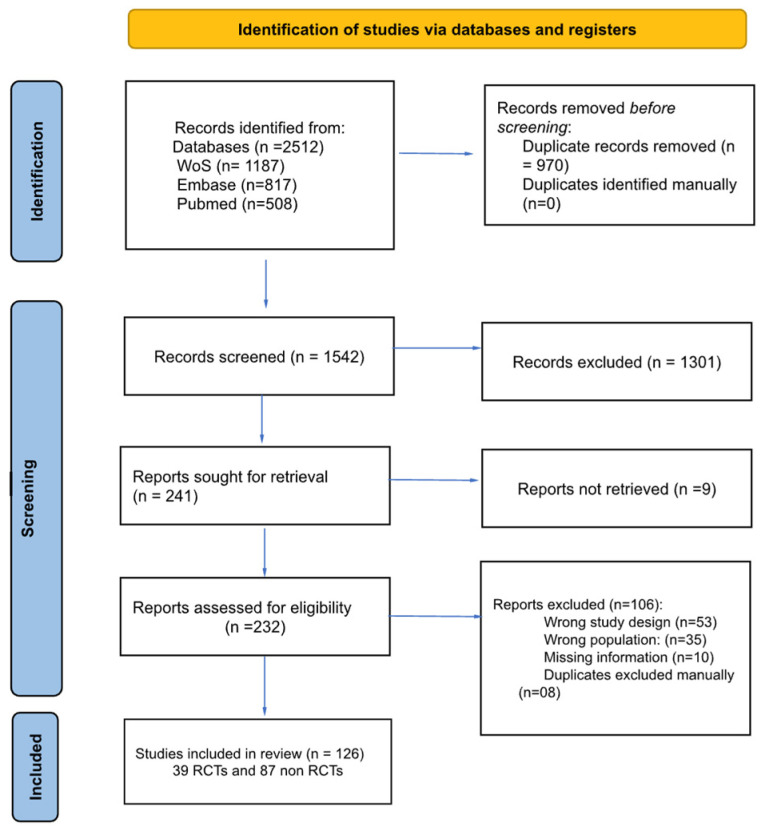
Study flow chart.

**Table 1 biomedicines-13-00988-t001:** Characterization of the studies using QST methods in patients with FMS from the retrieved studies (*n* = 126).

Author, Year	Design	ACR Diagnostic Criteria	Arms	Total Females (%)	Age Mean (SD)	QST Static	QST Dynamic
Sorensen et al., 1995 [26]	Non RCT	1990	3	100	47.5 (7.5)	Yes	No
Kosek et al., 1997 [27]	Non RCT	1990	2	100	42.5 (28.5)	Yes	No
Ernberg et al., 1997 [28]	Non RCT	1990	2	92	48.5 (14.3)	Yes	No
Hurtig et al., 2001 [16]	Non RCT	1990	2	100	43 (37.9)	Yes	No
Price et al., 2002 [29]	Non RCT	1990	2	100	45.5 (12.5)	No	Yes
Desmeules et al., 2003 [30]	Non RCT	1990	2	88	48.3 (10.3)	Yes	No
Staud et al., 2003 [31]	Non RCT	1990	1	87.3	49.9 (10.4)	No	Yes
Ernberg et al., 2003 [32]	Non RCT	1990	2	92	48.5 (14.3)	Yes	No
Kendall et al., 2003 [33]	Non RCT	1990	3	100	43.6 (7.3)	Yes	No
Yildiz et al., 2004 [34]	RCT	1990	2	70	40.1 (4.7)	Yes	No
Staud et al., 2004a [35]	Non RCT	1990	2	100	42.9 (13.04)	Yes	No
Staud et al., 2004b [36]	Non RCT	1990	1	94.6	49.6 (11.5)	Yes	No
Giesecke et al., 2005 [37]	Non RCT	1990	2	62.3	40.2 (9)	Yes	No
Montoya et al., 2005 [38]	Non RCT	1990	2	100	51.6 (5.9)	Yes	No
Staud et al., 2005 [39]	Non RCT	1990	2	100	47.05 (8.7)	Yes	No
Geisser et al., 2007 [40]	Non RCT	1990	3	100	39.6 (9.2)	Yes	No
Jespersen et al., 2007 [41]	Non RCT	1990	2	100	47 (6.08)	Yes	No
Smith et al., 2008 [42]	Non RCT	1990	2	100	48 (6.8)	Yes	No
Targino et al., 2008 [43]	RCT	1990	2	100	51.6 (11.07)	Yes	No
Diers et al., 2008 [44]	Non RCT	1990	2	86.7	50.4 (9.5)	Yes	No
Staud et al., 2008 [45]	Non RCT	1990	2	100	43.15 (9)	No	Yes
Suman et al., 2009 [46]	Non RCT	1990	2	100	44.8 (11.7)	Yes	No
Ge et al., 2009 [47]	Non RCT	1990	2	100	53 (2.4)	Yes	No
Stening et al., 2010 [48]	RCT	1990	2	100	54.3 (3.4)	Yes	No
Nelson et al., 2010 [49]	RCT	1990	2	100	51.7 (10.3)	Yes	No
Tastekin et al., 2010 [50]	Non RCT	1990	2	100	42.7 (6.7)	Yes	No
de Bruijn et al., 2011 [51]	Non RCT	1990	1	100	37.3 (7.7)	Yes	No
Hassett et al., 2012 [52]	Non RCT	1990	2	100	41.1 (10.8)	Yes	No
Martínez-Jauand et al., 2013 [53]	Non RCT	1990	2	100	50.5 (9.4)	Yes	No
Paul-Savoie et al., 2012 [54]	Non RCT	1990	2	100	49.8 (9.3)	Yes	Yes
Hargrove et al., 2012 [55]	RCT	1990	2	92.2	52.6 (3.1)	Yes	No
Hooten et al., 2012 [56]	RCT	1990	2	90.3	46.5 (10.8)	Yes	No
Hassett et al., 2020 [57]	Non RCT	1990	2	100	38.8 (11.7)	Yes	No
Castro-Sanchez et al., 2012 [58]	Non RCT	1990	2	50	52 (5.5)	Yes	No
Burgmer et al., 2012 [59]	Non RCT	1990	2	100	52.59 (7.95)	Yes	No
Van Oosterwijck et al., 2013 [60]	RCT	1990	2	80	45.8 (10.9)	Yes	No
Üçeyler et al., 2013 [61]	Non RCT	1990	2	91.42	56.4 (28.9)	Yes	No
Crettaz et al., 2013 [62]	Non RCT	1990	2	100	40.2 (9.2)	Yes	Yes
Da Silva et al., 2013 [15]	Non RCT	1990	2	96	49.9 (14.5)	Yes	Yes
Casanueva et al., 2013 [63]	RCT	1990	2	100	53.7 (10.8)	Yes	No
Belenguer-Prieto et al., 2013 [64]	Non RCT	1990	2	96.7	50.8 (7.8)	Yes	No
Staud et al., 2014a [65]	RCT	1990	3	100	45.8 (14.8)	Yes	No
Bokarewa et al., 2014 [66]	Non RCT	1990	3	100	51 (2.5)	Yes	No
Castro-Sanchez et al., 2014 [67]	RCT	1990	2	54	53.5 (7.5)	Yes	No
Staud et al., 2014b [68]	Non RCT	1990	2	100	45.8 (14.8)	Yes	No
Vandenbroucke et al., 2014 [69]	Non RCT	2010	2	94.8	39 (11.7)	Yes	No
Staud et al., 2015 [70]	RCT	1990	2	91.80	47.2 (12)	Yes	No
Qin et al., 2015 [71]	Non RCT	1990	2	86.05	45 (9.5)	Yes	No
Soriano-Maldonado et al., 2015 [72]	Non RCT	1990	1	100	48.3 (7.8)	Yes	No
Kim et al., 2015 [73]	Non RCT	2010	2	84	44.6 (13.08)	Yes	No
Zamuner et al., 2015 [74]	Non RCT	1990	2	100	47.07 (7)	Yes	No
Efrati et al., 2015 [75]	RCT	1990	2	100	49.2 (11)	Yes	No
Oudejans et al., 2016 [76]	RCT	1990	1	92.31	39.2 (60.1)	Yes	Yes
Potvin et al., 2016 [77]	Non RCT	1990	2	93.41	49.5 (8.2)	Yes	Yes
Schoen et al., 2016 [78]	Non RCT	1990	2	100	42.7 (10.8)	Yes	Yes
Barbero et al., 2017 [79]	Non RCT	1990/2010	1	100	49.5 (8.1)	Yes	No
Forti et al., 2016 [80]	Non RCT	1990	2	100	48.9 (7.2)	Yes	No
Gomez-Perretta et al., 2016 [81]	Non RCT	1990	2	100	46.2 (10.5)	Yes	No
Saral et al., 2016 [82]	RCT	1990	3	100	41.7 (7.7)	Yes	No
Mendonça et al., 2016 [83]	RCT	2010	3	97.8	19.5 (8.19)	Yes	No
Luciano et al., 2016 [84]	RCT	1990	1	100	57.28 (8.81)	Yes	No
Gerhardta et al., 2017 [85]	Non RCT	1990	3	72.88	56.8 (10)	Yes	Yes
de la Coba et al., 2017 [86]	Non RCT	1990	2	100	53.09 (9.38)	No	Yes
Freitas et al., 2017 [87]	Non RCT	1990	2	100	53.03 (10.2)	Yes	No
Baumueller et al., 2017 [88]	RCT	1990	2	100	55.6 (6.1)	Yes	No
Harper et al., 2018 [89]	Non RCT	1990	2	100	40.7 (11.2)	No	Yes
Pickering et al., 2018 [90]	RCT	2010	2	100	46.7 (10.6)	Yes	Yes
Merriwether et al., 2018 [91]	Non RCT	1990	1	100	49.3 (11.5)	Yes	Yes
Wodehouse et al., 2018 [92]	Non RCT	1990/2010	1	92.8	46.7 (10.5)	Yes	Yes
Albers et al., 2018 [93]	RCT	1990	3	100	55.4 (11.9)	Yes	No
Galvez-Sanchez et al., 2018 [94]	Non RCT	2010	2	100	49.02 (8.2)	Yes	No
Eken et al., 2018 [95]	Non RCT	1990	2	94	36.9 (7.5)	Yes	No
de la Coba et al., 2018 [96]	Non RCT	1990	2	100	53.09 (10.4)	No	Yes
Evdokimov et al., 2019 [97]	Non RCT	1990/2010	2	100	52 (15.8)	Yes	Yes
Brietzke et al., 2019 [98]	Non RCT	2010	2	100	42.2 (7.1)	Yes	Yes
Amer-Cuenca et al., 2020 [99]	RCT	1990/2010	4	100	53.2 (9)	Yes	Yes
Donk et al., 2019 [100]	RCT	2010	2	94.1	44.5 (22.6)	Yes	Yes
Andrade et al., 2019 [101]	RCT	1990	2	100	51.9 (8)	Yes	No
Udina-Cortés et al., 2020 [102]	RCT	2010	2	100	52 (8.8)	Yes	Yes
Uygur-Kucukseymena et al., 2020 [103]	Non RCT	2010	1	88.5	53 (13.52)	No	Yes
Kaziyama et al., 2020 [104]	Non RCT	2010	2	100	44.4 (6.3)	Yes	No
Pickering et al., 2020 [105]	Non RCT	2016	2	100	51 (9.6)	Yes	No
Sarmento et al., 2020 [106]	RCT	2010	2	100	48.8 (11.4)	Yes	No
Yuan et al., 2020 [107]	Non RCT	1990	2	97	51.07 (8.16)	Yes	No
Han et al., 2020 [108]	Non RCT	2010	2	97	52 (8.74)	Yes	No
Izquierdo-Alventosa et al., 2020 [109]	RCT	2016	2	100	54 (7.9)	Yes	No
Rehm et al., 2021 [110]	Non RCT	1990	1	95.5	50.4 (9.6)	Yes	No
Falaguera-Vera et al., 2020 [111]	Non RCT	1990/2010	2	100	55.6 (7.2)	Yes	No
Staud et al., 2021 [112]	Non RCT	1990	2	100	48 (11.9)	Yes	No
Jamison et al., 2022 [113]	RCT	2010	2	100	50.4 (13.5)	Yes	Yes
Jamison et al., 2021 [114]	RCT	2010	2	93.3	50.3 (13.5)	Yes	Yes
Soldatelli et al., 2021 [115]	Non RCT	2010/2016	2	100	49.3 (8.6)	Yes	Yes
Karamanlioglu et al., 2021 [116]	RCT	2010	2	100	43.7 (8.1)	Yes	No
Izquierdo-Alventosa et al., 2021 [117]	RCT	2016	3	100	52.8 (8.2)	Yes	No
Van Campen et al., 2021 [118]	Non RCT	2010	3	100	39.6 (12.3)	Yes	Yes
Weber et al., 2022 [119]	Non RCT	2016	2	81	49.9 (8.4)	Yes	Yes
Pacheco-Barrios et al., 2022 [120]	Non RCT	2010	1	86.21	47.6 (11.5)	Yes	Yes
De Paula et al., 2022 [121]	RCT	2016	4	100	49.3 (2.1)	Yes	Yes
Tour et al., 2022 [122]	Non RCT	1990	2	100	47.5 (7.8)	Yes	Yes
Serrano et al., 2022 [123]	Non RCT	2016	2	100	48.2 (9.8)	Yes	Yes
Alsouhibani et al., 2022 [124]	RCT	2010	2	88.4	49.8 (14.4)	Yes	Yes
Franco et al., 2022 [125]	Non RCT	2016	2	100	49.9 (10)	No	Yes
Samartin-Veiga et al., 2022 [126]	RCT	2010	4	100	50.2 (8.7)	Yes	No
Lin et al., 2022 [127]	RCT	2016	2	100	48.5 (13.02)	Yes	No
Castelo-Branco et al., 2022 [128]	Non RCT	2010	4	87.8	48.8 (10.1)	No	Yes
Berwick et al., 2022 [129]	Non RCT	1990/2010		90	49.4 (10.6)	Yes	No
Fanton et al., 2022 [130]	Non RCT	1990/2010		100	47.6 (7.7)	Yes	No
Ablin et al., 2023 [131]	RCT	2016	2	79.3	45.1 (12.3)	Yes	Yes
Berardi et al., 2021 [132]	RCT	1990	4	100	48.7 (11.7)	Yes	Yes
Cigaran-Mendez et al., 2023 [133]	Non RCT	1990/2010	2	100	52.5 (11)	Yes	Yes
Soldatelli et al., 2023 [115]	Non RCT	1990	2	100	49.6 (7.7)	No	Yes
Leone et al., 2023 [134]	Non RCT	2016	3	88.30	49.1 (11.7)	Yes	Yes
Bao et al., 2023 [135]	RCT	2016	3	100	43.6 (14.3)	No	Yes
Kumar et al., 2023 [136]	Non RCT	2010	1	100	35.1 (8.9)	Yes	No
Tapia-Haro et al., 2023 [137]	Non RCT	2010	1	100	56.06 (6.41)	Yes	No
Sanzo et al., 2024 [138]	RCT	2010	2	100	52.07 (2.28)	Yes	Yes
Baumler et al., 2024 [139]	Non RCT	2010/2016	2	100	54.9 (13.02)	Yes	No
Neira et al., 2024 [140]	Non RCT	1990/2010	2	100	50 (9)	Yes	No
Marshall et al., 2024 [141]	Non RCT	2016	3	93	45.4 (15.0)	Yes	No
Boussi-Gross et al., 2024 [142]	RCT	2016	2	100	33.3 (5.9)	Yes	Yes
Coupel et al., 2024 [143]	Non RCT	2010	2	98.2	50.91 (10.04)	Yes	No
Berardi et al., 2024 [144]	RCT	1990	4	100	49.05 (11.6)	Yes	Yes
Aoe et al., 2024 [145]	Non RCT	2016	2	90	42.4 (11.1)	Yes	No
Castelo-Branco et al., 2024 [146]	Non RCT	2010	1	86.5	48.08 (11.12)	No	Yes
Gil-Ugidos et al., 2024 [147]	Non RCT	2010	1	100	56.06 (6.41)	Yes	No
Gungormus et al., 2024 [148]	RCT	2016	2	100	54.5 (7.5)	Yes	Yes

American College of Rheumatology (ACR), Standard Deviation (SD), Quantitative Sensory Testing (QST).

**Table 2 biomedicines-13-00988-t002:** Summary of Static and Dynamic Quantitative Sensory Testing Across Body Locations.

Test Category	Testing Location	Test Category	Testing Location
Static Quantitative Sensory Testing		Dynamic Quantitative Sensory Testing	
Mechanical Detection, Pain threshold or Mechanical Pain Sensitivity	Forearm *n* = 3 Hands *n* = 4 Variable *n* = 3	Windup and Temporal Summation-Mechanical or Thermal	Forearm *n* = 4 Hands *n* = 7 Foots *n* = 1 Variable *n* = 3
Pressure Pain Threshold (PPT)	Forearm *n* = 5 Hands *n* = 17 Trapezius *n* = 8 18 tender points *n* = 24 Variable *n* = 30	Conditioned Pain Modulation (CPM)	Forearm *n* = 12 Hands *n* = 4 Foots *n* = 2 Variable *n* = 8
Cold Pain Threshold or Cold Pain Tolerance	Forearm *n* = 2 Hands *n* = 9 Variable *n* = 5		
Heat Pain Threshold or Tolerance	Forearm *n* = 7 Hands *n* = 6 Variable *n* = 7		

A summary of key findings from quantitative sensory testing, with a particular emphasis on the primary body locations targeted in these assessments. The left section of the table summarizes static sensory tests, such as mechanical and thermal detection/pain thresholds and pressure pain thresholds, while the right section focuses on dynamic sensory tests, including temporal summation and conditioned pain modulation. It should be noted that the referenced numbers correspond to studies that provide further detail on the test results for each specified location.

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
