# Peer review of "Quantitative Sensory Testing in Fibromyalgia Syndrome: A Scoping Review"

_biomedicines, 2025, doi:10.3390/biomedicines13040988_

Round 1
Reviewer 1 Report
Comments and Suggestions for Authors
Dear Authors
The subject of this study is relevant, however, its presentation must be strongly improved. Please, see in the attached file my considerations and suggestions.

Author Response
Re: Manuscript ID: biomedicines-3547230– "Quantitative Sensory Testing in Fibromyalgia Syndrome: A Scoping Review"
Dear Reviewer,
On behalf of all co-authors, I would like to express our sincere appreciation for your careful and insightful review of our manuscript. Your constructive comments were extremely valuable in improving the clarity, methodological rigor, and scientific contribution of our work.
We carefully considered each of your suggestions and have revised the manuscript accordingly. Below, we provide a detailed point-by-point response to all comments, indicating the modifications made in the revised version of the manuscript.
Comment (Line 38): “delete”
Response: We deleted the indicated content as requested.
Comment (Line 40): “I suggest adding more references. Some suggestions are…”
Response: We thank the reviewer for the valuable suggestions. Both references have been included in the revised manuscript to strengthen the contextualization of the topic. → Elli H, Akkus M (2025), and → Dos Santos JM et al. (2022).
Comment (Line 48): “add reference”
Response: The appropriate reference has been added to support the content in that section.
Comment (Line 65): Replace with: “..the current...”
Response: The suggested wording was implemented in the revised version of the manuscript.
Comment (Table 1): “Some studies of this Table are not among the publications indicated in the Reference section. Please, clarify.”
Response: Thank you for this observation. We reviewed all the references and ensured that each study listed in Table 1 is now included in the References section.
Comment (Table 1): “use...et al.. in all citations of this Table.”
Response: Adjustments were made throughout Table 1 to apply “et al.” uniformly, in accordance with the journal’s formatting and the reviewer’s suggestion.
Comment (Table 1): “I suggest using, Jamison, 2021a and Jamison, 2021b”
Response: We agree with this recommendation and have revised the references accordingly to clearly distinguish between the two studies published by Jamison in the same year.
Comment (Line 131): “delete”
Response: The indicated word was deleted as requested.
Comment (Line 175): “add references”
Response: Additional references have been included to support and reinforce the discussion in this section.
Comment (Line 187): “What are ‘these factors’? Clarify”
Response: Thank you for pointing this out. We rewrote the paragraph to improve clarity and specify the factors being referred to, ensuring that the discussion is more precise and informative.
We truly appreciate your insightful comments, which helped us enhance the clarity, consistency, and scientific rigor of our manuscript. We hope the revised version now meets the expectations of the reviewers and the journal.

Reviewer 2 Report
Comments and Suggestions for Authors PRISMA is used for systematic reviews and the title speaks of scoping reviews...this already gives you an idea of the poor methodological reliability
To me they are poorly reproducible and standardisable works
Good
Author Response
To Reviewer 2,
Re: Manuscript ID: biomedicines-3547230– "Quantitative Sensory Testing in Fibromyalgia Syndrome: A Scoping Review"
Dear Reviewer,
We thank you for reviewing our manuscript. We understand and acknowledge the concerns raised in your comments and have taken them seriously during the revision process.
Below, we present our point-by-point responses, providing clarification and justification for the methodological choices made in our scoping review. We hope our explanations and the revisions implemented in the manuscript adequately address your feedback.
Comment: PRISMA is used for systematic reviews and the title speaks of scoping reviews...this already gives you an idea of the poor methodological reliability
To me they are poorly reproducible and standardisable works
Response: Thank you for your comments.
Regarding this recommendation, it is important to highlight that we followed the PRISMA-ScR (Preferred Reporting Items for Systematic Reviews and Meta-Analyses extension for Scoping Reviews), which is the specific guideline designed for this type of review, as proposed by Tricco et al. (2018). This is clearly mentioned in the Methods section and cited accordingly in the references (Reference 19).
Regarding the concern that scoping reviews are poorly reproducible due to the lack of methodological reliability, we would like to highlight that our decision for a scooping is due the main proposal of this review. Scooping reviews also serve a valid and recognized purpose in synthesizing heterogeneous and complex fields of research, especially when mapping the extent and nature of evidence is necessary. In our case, the methodological diversity in Quantitative Sensory Testing (QST) for fibromyalgia is precisely why a scoping approach was chosen.
Moreover, we employed a structured protocol (registered on OSF), predefined eligibility criteria, independent screening by two reviewers using Rayyan, and risk of bias assessment tools (ROB 2 and STROBE) to ensure transparency and rigor throughout the review process.
We have revised the text to emphasize the rationale for choosing the scoping review methodology and the steps we took to ensure methodological reliability.
We hope this clarification helps address the reviewer’s concerns.

Reviewer 3 Report
Comments and Suggestions for Authors
This paper provides a very good scoping review of the use of the Quantitative Sensory Testing (QST) on Fibromyalgia Syndrome (FMS). It finds the lack of standardization in QST, i.e., in protocols, number of tender points, sites and types of tests and provides some ideas for future work to narrow down the study, e.g., separate studies by treatment modality (e.g., pharmacological vs. neuromodulatory). The authors did a massive work on literature analysis, both on RCT studies and non RCTs. However, the search of studies needs more clarification (see my specific comments below). The manuscript would benefit from adding a few sentences describing how QST is done (perhaps device is used?, photo could be added?, etc.), its output range, etc. to attract wider audience, see for example the recent discussions of the effect of pulsed electromagnetic field on patients with fibromyalgia. The included studies were divided by the static (76%) or dynamic (24%) QST, but the difference between the two QST methods were not clearly mentioned in the manuscript. The authors may want to add a couple sentences to clarify the difference between the two methods of QST. The authors also report in Table 1 the number of Arms reported in the studies (ranging from 1 to 4), but it is not explicitly explained in the manuscript; the reader may just guess that the arms could mean the experimental and control groups. So, the authors may want to clarify this by one or two sentences. Overall, good job was done, but there is room for improvement.
Line 72: should be Meta-Analyses, not Metanalysis
Line 85: “We replicated the primary database search terms (Pubmed) for the others (see Supplementary Material 2).” It is good that the authors submitted the supplementary file with the manuscript; however, the font size in Table of Supplementary 4 is too small, not clear why?
Line 87: “The search was made up to June 31, 2024.” However, June ends on 30th day, not 31. Also, in the supplementary file the date was set to: June 03, 2024. Which date did you use?
Line 108: “The search (up to June 2024) yielded 2512 records, of which 1542 remained after duplicate removal.” The search of publications in PubMed up to June 03, 2024 using the query from the authors (see Supplementary Material 2) yielded 513 citations. It is not clear why the authors had 508, because my search shows +5 articles. The authors do not mention about additional filters (e.g. language, article type, species, etc.) they may have used. The authors stated that they included " studies where participants were adults, both sexes, aged 18+". However, in PubMed, when all age filters starting with “Adult: 19+ years” included (i.e., no children “Child: birth-18 years”), the search yields 225 articles. So, it would be good to check and clarify the search query for PubMed first. For the reported n=508, I think that the authors did not use any additional filter.
My search in Web of Science using the query from the authors (see Supplementary Material 2) for articles published up to June 03, 2024 shows +8 articles. For the reported n=1187, I think that the authors did not use any additional filter. Also, in the query for WoS, there is typo: should be "allodynia", not allodyniaa".
The authors should also check the query for Embase, I think there are some typos in it.
Author Response
To Reviewer 3,
Re: Manuscript ID: biomedicines-3547230– "Quantitative Sensory Testing in Fibromyalgia Syndrome: A Scoping Review"
Dear Reviewer,
We would like to sincerely thank you for your detailed and constructive feedback on our manuscript. Your careful evaluation allowed us to correct typographical inconsistencies, improve the clarity of our methodology, and enhance the overall quality of the submission. Below, we provide a point-by-point response to each of your comments and describe the corresponding revisions made.
Comment: “Line 72: should be Meta-Analyses, not Metanalysis”
Response: Thank you for the correction. The term has been revised to "Meta-Analyses" as suggested.
Comment: “Line 85: [...] the font size in Table of Supplementary 4 is too small…”
Response: We appreciate you pointing this out. We have revised Supplementary Table 4 and adjusted the font size to ensure proper readability across devices and formats.
Comment: “Line 87: 'The search was made up to June 31, 2024.' However, June ends on 30th day, not 31. Also, in the supplementary file the date was set to: June 03, 2024. Which date did you use?”
Response: Thank you for catching this error. Indeed, this was a typographical mistake. The correct date for the final search was June 3rd, 2024, as indicated in Supplementary Material 2. We have corrected this information in the main text accordingly.
Comment: “Line 108: [...] the query from the authors yielded 513 citations, but authors report 508 [...] authors do not mention additional filters [...] Also, authors stated that participants were adults [...] but when filtered for 19+ years, PubMed shows 225 articles...”
Response: We greatly appreciate your thorough verification. The discrepancy may be due to minor differences in the database's real-time updates. We confirm that no automatic filters (such as age, article type, or species) were applied via the PubMed interface. Instead, eligibility criteria (e.g., adults ≥18 years, original studies) were applied manually during the
screening phase after exporting the raw results. We have clarified this process in the Methods section to improve transparency.
Comment: “My search in Web of Science using the query from the authors [...] shows +8 articles [...] typo: should be 'allodynia', not allodyniaa”
Response: Thank you for pointing out the typographical error in the Web of Science search string ("allodyniaa" instead of "allodynia"). We confirm the typo was present in the documentation of the search strategy (Supplementary Material 2), but not in the actual search performed during the review process. The search was initially run with the correct term "allodynia," and thus the number of retrieved records (n = 1187) reflects the intended search query. We have corrected the typographical error in the documentation, and this update has been made in the revised Supplementary Material 2. No changes were made to the study selection or flowchart, as the search results remain unchanged.
Comment: “The authors should also check the query for Embase, I think there are some typos in it.”
Response: Thank you for this observation. We carefully reviewed the Embase search strategy and identified minor typographical errors, which have now been corrected in Supplementary Material 2.
We truly appreciate your insightful comments, which helped us enhance the clarity, consistency, and scientific rigor of our manuscript. We hope the revised version now meets the expectations of the reviewers and the journal.

Round 2
Reviewer 1 Report
Comments and Suggestions for Authors
Congratulations. The presentation of the manuscript was improved, and I am recommending its acceptance.
Author Response
Thank you very much for your kind words and for recommending the acceptance of our manuscript. We are truly grateful for your feedback and support throughout the revision process.
Best regards,
Valquiria A. Silva
On behalf of all co-authors
Reviewer 2 Report
Comments and Suggestions for Authors
the revision is ok for me
Author Response
Dear Reviewer,
Thank you for taking the time to review our manuscript and for your approval of the revision. We appreciate your contribution to improving our work.
Best regards,
Valquiria A. Silva
On behalf of all co-authors
Reviewer 3 Report
Comments and Suggestions for Authors
I think the authors addressed most of my comments. A couple of minor comments are below:
Figure 1, should be RCTs, not RCTS. Perhaps, WoS not WOS.
In the query for WoS, there is still typo: I think it should be "allodynia", i.e., " is missing before allodynia".
Author Response
Dear Reviewer,
Thank you again for your careful review and helpful comments. We have corrected the issues you pointed out:
-
“RCTS” was changed to “RCTs” in Figure 1.
-
“WOS” was corrected to “WoS”.
-
We also added the missing quotation mark before “allodynia” in the WoS search query.
We appreciate your attention to detail.
Best regards,
Valquiria A Silva
On behalf of all co-authors
Round 3
Reviewer 3 Report
Comments and Suggestions for Authors
Excellent work!
I do not have any other comments.